# ROS- and pH-Responsive Polydopamine Functionalized Ti_3_C_2_T_x_ MXene-Based Nanoparticles as Drug Delivery Nanocarriers with High Antibacterial Activity

**DOI:** 10.3390/nano12244392

**Published:** 2022-12-09

**Authors:** Wei-Jin Zhang, Shuwei Li, Veena Vijayan, Jun Seok Lee, Sung Soo Park, Xiuguo Cui, Ildoo Chung, Jaejun Lee, Suk-kyun Ahn, Jung Rae Kim, In-Kyu Park, Chang-Sik Ha

**Affiliations:** 1Department of Polymer Science and Engineering, School of Chemical Engineering, Pusan National University, Busan 46241, Republic of Korea; 2School of Chemical Engineering, Pusan National University, Busan 46241, Republic of Korea; 3Department of Biomedical Sciences, Chonnam National University Medical School, Gwangju 61469, Republic of Korea; 4BioMedical Sciences Graduate Program (BMSGP), Chonnam National University, Hwasun 58128, Republic of Korea; 5Division of Advanced Materials Engineering, Dong-Eui University, Busan 47340, Republic of Korea; 6School of Material Science and Engineering, Beijing Institute of Petrochemical Technology, Beijing 102617, China

**Keywords:** MXene (Ti_3_C_2_), ROS/pH-responsive, photothermal conversion property, antibacterial activity, drug delivery

## Abstract

Premature drug release and poor controllability is a challenge in the practical application of tumor therapy, which may lead to poor chemotherapy efficacy and severe adverse effects. In this study, a reactive oxygen species (ROS)-cleavable nanoparticle system (MXene-TK-DOX@PDA) was designed for effective chemotherapy drug delivery and antibacterial applications. Doxorubicin (DOX) was conjugated to the surface of (3-aminopropyl)triethoxysilane (APTES)-functionalized MXene via an ROS-cleavable diacetoxyl thioketal (TK) linkage. Subsequently, the surfaces of the MXene nanosheets were coated with pH-responsive polydopamine (PDA) as a gatekeeper. PDA endowed the MXene-TK-DOX@PDA nanoparticles with superior biocompatibility and stability. The MXene-TK-DOX@PDA nanoparticles had an ultrathin planar structure and a small lateral size of approximately 180 nm. The as-synthesized nanoparticles demonstrated outstanding photothermal conversion efficiency, superior photothermal stability, and a remarkable extinction coefficient (23.3 L g^−1^ cm^−1^ at 808 nm). DOX exhibited both efficient ROS-responsive and pH-responsive release performance from MXene-TK-DOX@PDA nanoparticles due to the cleavage of the thioketal linker. In addition, MXene-TK-DOX@PDA nanoparticles displayed high antibacterial activity against both Gram-negative *Escherichia coli* (*E. coli*) and Gram-positive *Bacillus subtilis* (*B. subtilis*) within 5 h. Taken together, we hope that MXene-TK-DOX@PDA nanoparticles will enrich the drug delivery system and significantly expand their applications in the biomedical field

## 1. Introduction

MXene (Ti_3_C_2_T_x_), an emerging family of graphene analog two-dimensional nanomaterials, has garnered considerable attention because of its typical planar topology and intriguing physiochemical performance since it was first proposed by Gogotsi et al. [1]. This can be achieved by selectively etching the Al atomic layer of Ti_3_AlC_2_, where M represents the transition metal, A represents an element from group 13 or 14 in the periodic table, X represents C or N, and T represents the terminated groups (-O, -OH, and/or-F) of the MXene surface [2]. To date, over 70 types of MXenes have been proposed and synthesized with various elements, and more than 100 have been theoretically predicted to exhibit excellent conductivity, good mechanical stability, and high biocompatibility. MXenes have been used in a variety of fields including batteries [3,4,5], electromagnetic interference (EMI) shielding [6], energy storage [7], antibacterial applications [8], and catalysis [9]. Wan et al. [10] successfully fabricated Ti_3_C_2_T_x_ MXene/bacterial cellulose (BC) composite films with ultrathin, strong, and highly flexible properties via in situ biosynthesis. However, there are few reports on MXene-based nanoplatforms for biomedical applications including photoacoustic (PA) imaging, antimicrobial activity, biosensors, and anticancer therapy. There is an urgent need to explore the cellular and animal levels as well as other bioscience applications.

Doxorubicin (DOX) is one of the most effective chemotherapeutic agents and is used for the treatment of various cancers by acting on DNA [11]. Chemotherapy is one of the most significant and efficient clinical therapies for cancer treatment. Nevertheless, the high toxicity of chemotherapeutic drugs to normal cells is a major obstacle in clinical practice [12]. Therefore, many novel nanotechnologies have been proposed and designed because of their huge potential for increasing therapy efficiency and reducing toxicity. Yao et al. [13] reported a novel near-infrared(NIR) light-controlled drug-release device based on composites of silica@polypyrrole@mesoporous silica (SiO_2_@PPy@mSiO_2_) and triblock poly(ethylene glycol)-poly(ε-caprolactone)-poly(ethylene glycol) (PEG-PCL-PEG) polymers. Zhang et al. created a ZnO@mesoporous silica nanoparticle (MSN) nanocarrier [14] by covering the decalysine sequence (K10) and reacting it with citraconic anhydride. Subsequently, zinc oxide (ZnO) quantum dots (QDs) were added to cap MSNs using electrostatic contact. The toxicity of a drug can be reduced by encapsulating it in the pores or channels of a delivery carrier via physical means. Nevertheless, the unexpected release, lack of control of tumor locoregional drug release, and poor tumor-homing of nanomedical devices severely restrict their clinical application, which is attributed to the large and open channels of nanocarriers. In addition, premature drug release poses serious risks to patients including hematotoxicity, nephrotoxicity, hepatotoxicity, and immunogenicity [15,16]. Hence, a drug delivery system with “zero-premature” drug release and stimuli-responsive controlled drug release characteristics is urgently needed in practice. Because of these drawbacks, various gatekeepers such as β-cyclodextrin [17], PEG [18], and folic acid [19] have been used to overcome these issues, which can be introduced by stimulus-responsive (pH-, magnetic-, reactive oxygen species (ROS)-, humidity-) cross-linkers [20]. Among the stimuli-responsive cross-linkers, the sulfhydryl-assisted cleavage strategy is the most widely used. Diacetoxyl thioketal, another sulfhydryl-based ROS-responsive bond, has often been used for ROS-sensitive delivery [21]. As is well-known, ROS is one of the hallmarks of tumor tissue. For instance, hypoxia significantly alters the ROS levels in cancer cells. The ROS levels in tumor tissues (up to 100 × 10^−6^ M) are much higher than in normal tissue (≈20 × 10^−9^ M) [22] due to increased metabolic activity and oncogenic transformation [23]. It has also been reported that MXene sheets are capable of producing ROS, which can further increase the ROS levels in the tumor sites [24]. Furthermore, considering the induction of ROS by photodynamic therapy (PDT), ROS-responsive nanocarriers are uniquely suited for combining PDT with other therapies [25]. It should be noted that chemotherapeutic agents are coupled with the drug delivery system through covalent bonds, which can minimize DOX leakage during systemic circulation, reduce the in vivo fate of multiple drugs, and improve drug release at the locoregional of tumors [23,26].

It is well-known that MXene (Ti_3_C_2_) shows a certain photothermal conversion effect due to the absorption of NIR light and specific surface activity [27]. However, for tumor therapy, MXene only has a single photothermal capacity and is relatively deficient in killing cancer cells. Therefore, it is necessary to endow MXenes with additional photothermal conversion capabilities for effective cancer therapies. Polydopamine (PDA), a mussel-inspired biomaterial, is well-known for its salient antibacterial properties, low cytotoxicity, good biocompatibility, notable photothermal conversion properties, pH-responsiveness, and strong adhesive properties [28,29], making it a promising candidate for biomaterials. Lynge et al. [30] proved that the cargo uptake efficiency depends on the thickness of the PDA capping layer, which indicates that the proper PDA coating helps cellular uptake. It is well-known that the cellular uptake analysis of nanoparticles is indispensable in clinical practice, and some nanomedicines have even entered clinical trials. Zhang et al. [31] claimed that MXene (Ti_3_C_2_)-based quantum dots (Ti_3_C_2_-QDs) are localized around mitochondria through endocytosis and diffusion effects. Cao et al. [32] reported on vanadium carbide quantum dots (V_2_C QDs) photothermal agents and an engineered exosome (Ex) vector (V_2_C-TAT@Ex) and investigated their nucleus uptake ability by confocal laser scanning microscope (CLSM) images, showing the endocytic uptake pathway of V_2_C-TAT@Ex. Unal et al. [33] explored the antiviral properties and immune-profile of a large panel of four highly stable and well-characterized MXenes-Ti_3_C_2_T_x_, Ta_4_C_3_T*_x_*, Mo_2_Ti_2_C_3_T*_x_*, and Nb_4_C_3_T*_x_*, taking SARS-CoV-2 as a model. This work also mentioned that Ti_3_C_2_T_x_ significantly inhibited the production of IL-2 and IL-4, which have also been involved in COVID-19 disease severity, and their inhibition through JAK1/JAK3 blockade is currently being tested in phase 2 clinical trials (NCT04332042). The effect of the physicochemical properties of this carrier on cellular uptake will be implemented in our next work. Furthermore, because dopamine has catechol and amine functional groups, it can be easily formed by spontaneous polymerization in a pH 8.5 tris(hydroxymethyl) aminomethane water solution at room temperature, which can tightly adhere to the surface of the substance [34]. It is worth noting that polydopamine not only improves the biostability, but also enhances the photothermal conversion property of the nanodevice.

Given the facts described above, we designed a ROS- and pH-responsive compounding system (MXene-TK-DOX@PDA nanoparticles) with improved efficiency for chemodynamic therapy and reduced premature release toxicity, in which 3-aminopropyltriethoxysilane (APTES)-functionalized MXene (HF etching) and a chemotherapeutic agent, DOX, were conjugated to an identical diacetoxyl thioketal (TK) linker via a facile method. Subsequently, pH-responsive polydopamine was conjugated on the surface of MXene-based nanoplatforms to increase the photothermal conversion capacity and promote physiological stability. In this regard, the chemotherapeutic drug DOX was coupled with MXene-based nanovectors via covalent bonds, showing many merits including, but not limited to, avoiding the premature release of drugs, enhancing the loading capacity, and reducing the adverse effects. The MXene-TK-DOX@PDA nanoparticles demonstrated notable ROS- and pH-responsive behavior in vitro and provided a wealth of opportunities to obtain better outcomes. The photothermal conversion results of the MXene-TK-DOX@PDA nanoparticles indicate that the compounding system possesses sufficient ability to ablate cancer cells in practice. In addition, the antibacterial data revealed the high growth inhibition abilities of both *E. coli* and *B. subtilis* at a concentration of 6.085 mg/mL.

## 2. Experimental

### 2.1. Materials

Ti_3_AlC_2_ powder (MAX phase, ≥99%, ~400 mesh) was purchased from Ji-Lin Yiyi Technology Co. Ltd. (Jilin, China). 3-Aminopropyltriethoxysilane (APTES) (98%), acetone (anhydrous), thioglycolic acid (99%), toluene (anhydrous), hydrofluoric acid (48%), doxorubicin hydrochloride (99% HPLC), dimethyl sulfoxide (anhydrous 99.9%, DMSO), N-hydroxysuccinimide (NHS, 98.0%), hydrogen peroxide (30%, H_2_O_2_), tris(hydroxymethyl) aminomethane (>99.8%), trifluoroacetic acid (TFA, 99%), and dopamine hydrochloride were all provided by Sigma-Aldrich (Saint Louis, MO, USA). 1-(3-Dimethylaminopropyl)-3-ethylcarbodiimide hydrochloride (EDC, 98.0%) was supplied by TCI (Tokyo, Japan). Tetrabutylammonium hydroxide (TBAOH) was supplied by Alfa Aesar (Ward Hill, MA, USA). Phosphate buffered saline (PBS) was obtained from Welgene (Gyeongsan, Republic of Korea). Ethyl alcohol (94%) and n-hexane (95%) were supplied by Samchun Pure Chemicals Company (Seoul, Republic of Korea). Ethyl acetate (extra pure, 99%) was purchased from Junsei (Tokyo, Japan). The deionized water used in all of experiments was prepared using a Direct-Q^®^3 water purification system (EMD Millipore). All of the reagents were used without further purification.

### 2.2. Synthesis of MXene Nanosheets

Initially, the MAX-phase Ti_3_AlC_2_ was dispersed into a 20 mL 40% aqueous HF [35] solution in a Teflon bottle and stirred for 30 min to form a homogenous solution. Subsequently, 1 g of Ti_3_AlC_2_ powder was slowly added to a Teflon bottle over 30 min. It is worth noting that adding the MAX-phase too fast may cause sputtering of the HF aqueous solution because a large amount of hydrogen gas and heat could be generated. The reaction was performed in an oil bath at 35 °C for 24 h. Subsequently, the Al atomic layer was selectively etched after 24 h. To remove excess HF, the solution was washed with deionized water by centrifugation until a neutral supernatant solution was obtained. The product (Ti_3_C_2_) was collected by centrifugation at 3500 rpm for 30 min and freeze-dried for two days. One gram of the collected precipitate was exfoliated using 24 mL 25% TBAOH at 25 °C for 4 h. The products thus obtained were purified with ethanol and water several times for the removal of excess TBAOH. Pure MXene was obtained by centrifugation at 3500 rpm for 30 min, and the high-concentration supernatant was collected for future use. The procedure for the preparation of Ti_3_C_2_ is shown in Figure 1a.

### 2.3. Synthesis of MXene-NH_2_ Nanoparticles

Pure MXene (1 g) was added to toluene (60 mL) and rapidly stirred for 60 min under a nitrogen atmosphere at room temperature. Subsequently, 2 mL of APTES was added to the above solution, which was heated to 105 °C in a condensate reflux tube for 8 h [36]. MXene-NH_2_ was purified using deionized water and ethyl alcohol several times to remove excess APTES. The product was dried by lyophilization for 48 h.

### 2.4. Synthesis of MXene-TK Nanoparticles

To introduce the ROS-rupturable linker diacetoxyl thioketal, a DMSO solution of MXene-NH_2_ (50 mL, 2 mg/mL) was added to a DMSO solution of diacetoxyl thioketal (50 mL, 2 mg/mL) containing 1-(3-dimethylaminopropyl)-3-ethylcarbodiimide hydrochloride (0.6 mL) and N-hydroxysuccinimide (0.5 mL) for 48 h at 25 °C. The MXene-TK nanosheets were then purified with distilled water six times and freeze-dried for 48 h [37].

### 2.5. Synthesis of MXene-TK-DOX Nanosized Flakes

Doxorubicin hydrochloride (30 mg) was dispersed in DMSO (2 mL) solution containing 50 μL of triethylamine under an N_2_ environment overnight. MXene-TK (30 mg) was added to DMSO (15 mL) containing 30 mg of EDC and 20 mg of NHS as the active agent and activated for 1 h. Subsequently, the DOX/DMSO solution was injected into the above solution and reacted for another 72 h at 30 °C. The products were washed with water several times by centrifugation (4000 rpm for 5 min) and dried by lyophilization for 48 h.

### 2.6. Preparation of MXene-TK-DOX@PDA Nanoparticles

An aqueous solution of tris(hydroxymethyl)aminomethane (100 mL, pH 8.5) was prepared from tris(hydroxymethyl)aminomethane (0.606 g), HCl, and H_2_O (100 mL). Subsequently, 0.37 g MXene-TK-DOX was dispersed in the solution and stirred for 1 h to obtain a homogenous solution. Subsequently, 0.0956 g of dopamine hydrochloride was added to the above mixture and stirred for 24 h at 25 °C [38]. Excess chemicals were removed using water and dried by freeze-drying. It should be noted that the self-polymerization of dopamine requires an air environment. The preparation process of MXene−TK−DOX@PDA nanoparticles is illustrated in Figure 1b.

### 2.7. Synthesis of Diacetoxyl Thioketal (TK) Linker

The preparation process for diacetoxyl thioketal (TK) is shown in Appendix A. Thioglycolic acid (11.4 g), acetone (2.9 g), and trifluoroacetic acid (1.13 g) were reacted at room temperature for 24 h [39]. Subsequently, the reaction was quenched by adding ethyl acetate and the product was precipitated. Subsequently, white powder was obtained by washing with hexane and water to remove the excess reactants and dried in a vacuum oven at 50 °C for 24 h. It is worth noting that this reaction is an exothermal reaction and may appear as an agglomeration phenomenon. This problem can be solved by adding more acetone solvent. In addition, the structural information of the thioketal linker was evaluated by ^1^H- and ^13^C- nuclear magnetic resonance (NMR) spectroscopy (Appendix A, b) and FTIR spectroscopy, as depicted in Appendix A. ^1^H NMR (600 MHz, DMSO) δ 12.62 (s, 1H), 3.37 (s, 2H), 1.54 (s, 3H). ^13^C NMR (600 MHz, DMSO) δ 171.85 (a; HOO**C**-CH_2_-), δ 33.26 (b; -**C**H_2_-), δ 30.57 (c; -**C**H_3_), δ 56.69 (d; -**C**-).

### 2.8. Antibacterial Activity

The bacteriostatic performance of the products was evaluated using the standard colony-counting method. At 30 °C, the microbes were activated in Luria-Bertani (LB) medium. The microbial suspension was diluted 200-fold and redistributed in an equal volume of 0.01 mM phosphate buffer solution/product (such as MXene or MXene-TK-DOX or MXene-TK-DOX@PDA). The microbial dilution liquid (0 h, 5 h) was then spread evenly on the LB solid medium and cultured at 30 °C for 10 h. In addition, the viable number of microbial colonies was calculated by visual observation. All sample analyses were performed in triplicate.

### 2.9. Loading Capacity and Drug Release Study

MXene-TK-DOX@PDA nanoparticles (10 mg) were dispersed in H_2_O_2_/PBS (1.5/1.0 *v*./*v*.) solutions (pH 5.5, 5 mL), and the solution was shaken at 37 °C for 12 h. The supernatant was collected by centrifugation for UV measurement at 495 nm.

The loading capacity (LC) and encapsulation efficiency (EE) were calculated using the following equations [40]:(1)$$\mathrm{LC}\;(\%)=\frac{\mathrm{Weight}\;\mathrm{of}\;\mathrm{loaded}\;\mathrm{DOX}\;\mathrm{in}\;\mathrm{nanoparticles}\;}{\mathrm{Weight}\;\mathrm{of}\;\mathrm{nanoparticles}}\times 100,$$
(2)$$\mathrm{EE}\;(\%)=\frac{\;\mathrm{Weight}\;\mathrm{of}\;\mathrm{loaded}\;\mathrm{DOX}\;\mathrm{in}\;\mathrm{nanoparticles}}{\mathrm{Weight}\;\mathrm{of}\;\mathrm{DOX}\;\mathrm{in}\;\mathrm{feed}}\times 100.$$

The drug release profiles of the MXene-TK-DOX@PDA nanoparticles were analyzed in diverse media on a shaker (120 rpm) at 37 °C. In brief, MXene-TK-DOX@PDA nanoparticles were added to dialysis tubing with a molecular weight cut-off of 12 kDa. Subsequently, dialysis sacks were immersed in the medium. At a predetermined time, 3 mL of the medium was removed and replenished with the same volume of fresh medium. The absorbance of doxorubicin was detected by UV–Vis spectroscopy at 495 nm.

The cumulative drug release percentage was calculated using the following equation [41]:% R_t_ =C_t_·V_1_ + V_2_· (C _t−1_ + C _t−2_ + · · · + C_0_)/W_0_·L × 100%,(3)
where C_t_ denotes the drug concentration at time interval t; C_t−1_ and C_t−2_ are the drug concentrations prior to time interval t (C_0_ = 0); V_1_ is the total volume; and V_2_ is the volume extracted. W_0_ represents the initial weight of MXene-TK-DOX@PDA and L is the loading content.

### 2.10. Photothermal Conversion Property

The photothermal profile of the MXene-TK-DOX@PDA samples under an 808 nm laser was recorded using a thermal camera. Different concentrations of MXene-TK-DOX@PDA were dissolved in water and irradiated with an 808 nm laser at 2 W/cm^−2^ for 10 min. The temperature was recorded every 60 s and plotted. The photothermal profile of 300 µg mL^−1^ under different laser power densities (0.5, 1, 1.5, and 2 W cm^−2^) was also recorded and plotted. Furthermore, the photothermal stability of the samples was analyzed by five cycles of the ON/OFF laser with 5 min of irradiation for each cycle.

### 2.11. Characterization

The chemical structure of the thioketal was determined by nuclear magnetic resonance (NMR) spectroscopy (Bruker (600 MHz)) (Bruker Co., Billerica, MA, USA) using DMSO-d_6_ as the solvent. Fourier transform infrared (FTIR) spectra (JASCO FTIR 4100) (JASCO Co. Tokyo, Japan) were obtained using KBr pellets in the range of 4000–400 cm^−1^. The morphology of the products was analyzed at an accelerating voltage of 200 kV using high-resolution transmission electron microscopy (HR-TEM; JEOL 2010) (JEOL Ltd., Tokyo, Japan). X-ray diffraction (XRD, Bruker AXS) (Billerica, MA, USA) was carried out by Cu-Kα radiation over a wide range from 5° to 80° 2θ. The thermal stability of the samples was estimated by thermogravimetric analysis (TGA, Q50 V6.2, Build 187, TA Instruments, New Castle, DE, USA) in an N_2_ environment from 30 to 650 °C at a heating rate of 10 °C/min. The surface charge of the nanomaterials was evaluated using a zeta potential analyzer (Zetasizer, Malvern, UK). The hydrodynamic diameter of the products was measured using dynamic light scattering (DLS, Zetasizer NANO-S90, Malvern, UK). Raman spectra were obtained using confocal Raman spectroscopy (JASCO, NRS-5000 Series, Tokyo, Japan) with a visible laser at 532 nm. The diameter and thickness of the MXene were tested using atomic force microscopy (AFM; Park NX10) (Park System, Suwon, Republic of Korea). The morphology of the samples was examined using field emission scanning electron microscopy (FE-SEM, JEOL 6400) (JEOL Ltd., Tokyo, Japan) at an operating voltage of 20 KV with energy dispersive X-ray analysis (EDX). Chemical composition analysis of the MXene-TK-DOX@PDA nanoparticles and MXene was performed using X-ray photoelectron spectroscopy (XPS, VG Scientific (Cardiff, UK), Multi Lab) with Al Kα radiation. Infrared thermal images were captured and assayed using a thermal camera (Avio IR Camera/Thermometer, Tokyo, Japan). The drug release profile of the MXene-TK-DOX@PDA nanoparticles was determined via ultraviolet–visible (UV–Vis) spectrophotometry (Hitachi U-2010) (Kyoto, Japan).

## 3. Results and Discussion

### 3.1. Design, Preparation, and Characterization of MXene-Based Drug Delivery Nanoparticles

Ultrathin planar MXene (Ti_3_C_2_) nanosized flakes were obtained via HF etching and tetrabutylammonium hydroxide (TBAOH) delamination (Figure 1a), which showed high dispersity and desirable histocompatibility to meet the strict requirements of biomedical applications [42,43]. Initially, the Al atomic layer was removed by dispersing Ti_3_AlC_2_ in a hydrofluoric acid solution and stirring in a 35 °C oil bath. Although the HF etching process could exfoliate the Al atomic layer, the products still exhibited a bulky structure with a large size, which makes them difficult to apply in the biomedical field. To obtain monolayer or few-layer nanosheets, a delamination process with TBAOH is required, which can diffuse and intercalate into multilayer MXene.

The four-step preparation process for the dual (pH- and ROS-) responsive MXene-TK-DOX@PDA nanoplatforms is shown in Figure 1b. First, the delaminated MXene was modified with the organic silicon, APTES, via hydrolysis to introduce an amino group into the surface of MXene. Second, the ROS-cleavable linker was reacted with APTES-functionalized MXene via condensation polymerization between the carboxyl group of TK and the amino group of APTES with EDC and NHS as the cross-linker. Third, the chemotherapeutic agent DOX was conjugated to the surface of the functionalized MXene via condensation polymerization. Finally, pH-responsive polydopamine was introduced via dopamine self-polymerization at pH 8.5 to improve the biocompatibility and stability. Notably, polydopamine is a pH-sensitive biomaterial with salient photothermal conversion properties. The TEM image (Figure 1c) depicts the typical stacked layer structure, possessing a large lateral size of MXene, after HF etching. However, Figure 1d shows a highly dispersed and ultrathin flake morphology with a small lateral size (~200 nm) after intercalation by TBAOH. After delamination by TBAOH, the (002) peak showed a shift to a lower 2*Ɵ* angle from 8.1° (d = 1.06 nm) (as marked in yellow arrow in Figure 1e) to 5.7° (d = 1.54 nm) (as marked in yellow arrow in Figure 1f). These results further prove the successful removal of the Al atomic layer and intercalation of TBAOH into MXene [44,45]. The morphology of the MXene-TK-DOX@PDA nanoparticles was also evaluated using TEM. As shown in Figure 1g, the lateral size and morphology did not change significantly after modification with APTES, TK, or DOX. Additionally, a distinct organic layer, marked by the red arrow, was observed in the TEM image, and probably corresponded to the pH-responsive polydopamine. This result further demonstrates the successful coating of polydopamine on the MXene surface. The crystalline lattice of the MXene nanosheets is shown in Figure 1h, which clearly reveals the hexagonal structure and indicates the highly crystalline characteristics of MXene flakes [46]. The optical image (Figure 1i) illustrates the synthesis process of the MXene nanosheets and the DOX release profile from the MXene-TK-DOX@PDA nanoparticles.

### 3.2. Physicochemical Analysis

The wide-angle powder XRD patterns of the commercial Ti_3_AlC_2_ powder, MXene, and MXene + TBAOH nanosheets are presented in Figure 2a. The characteristic Bragg diffraction peaks of the Ti_3_AlC_2_ powder were observed at (002), (004), (101), (103), (104), (105), (107), (108), (109), and (110), which were associated with the 2θ values of 9.36°, 19.14°, 34.04°, 36.66°, 38.68°, 41.57°, 48.29°, 52.12°, 56.41°, and 60.16°, respectively. After etching, the characteristic (104) peak at 2θ = 39° that belongs to Ti_3_AlC_2_ powder disappeared, whereas the (002) peak shifted to a lower 2*Ɵ* angle from 9.5° (d = 0.92 nm) to 8.1° (d = 1.06 nm) [47]. After delamination with TBAOH, the (002) peak further shifted to a lower 2θ angle from 8.1° (d = 1.06 nm) to 5.7° (d = 1.54 nm). These results demonstrate that MXene was successfully etched and that TBAOH was successfully inserted into MXene.

The thermal stabilities of Ti_3_AlC_2_, MXene, MXene-NH_2_, MXene-TK, MXene-TK-DOX, and MXene-TK-DOX@PDA were estimated using thermogravimetric analysis (TGA) in the range of 30–650 °C in a N_2_ environment (Figure 2b). Ti_3_AlC_2_ exhibited a negligible weight loss over the entire temperature range up to 650 °C, demonstrating its high thermal stability (MAX phase) in the presence of N_2_. Notably, the weight of Ti_3_AlC_2_ increased slightly at high temperatures, which can be plausibly associated with the trace oxygen in nitrogen, and oxidation of the Al atomic layer in the MAX phase may occur [48]. The MXene nanosheets showed a weight loss of 1.7% below 200 °C, which was attributed to the evaporation of the physically absorbed water [49]. Subsequently, the mass of MXene was further reduced (6.1%) because decomposition of the functional groups (-OH, -F, and =O) may possibly occur at the surface of MXene. After modification with APTES, a 15% mass loss was observed for MXene-NH_2_ because of the decomposition of functional groups in APTES, which indicates the successful grafting of APTES onto the MXene surface. Additionally, the residual weights of the MXene-TK, MXene-TK-DOX, and MXene-TK-DOX@PDA nanoparticles were 82.1%, 69.3%, and 56.8%, respectively, after a heat treatment at temperatures as high as 650 °C at a rate of 10 °C/min in a nitrogen environment. The successful immobilization of TK, DOX, and polydopamine on the MXene surface was confirmed by the observed weight loss. These results demonstrate the successful fabrication of MXene-TK-DOX@PDA nanoparticles.

These findings are further supported by the zeta potential analysis, as shown in Figure 2c. The pristine MXene nanosheets exhibited a negatively charged surface with a zeta potential of −39.6 mV due to the negatively charged hydroxyl species terminated on the MXene surfaces [50], which was confirmed using XPS analysis, as described in a later section. After grafting the APTES, the zeta potential of MXene-NH_2_ was 12.7 mV, which showed a reversed charge due to APTES containing the amino group. Furthermore, the zeta potentials of the MXene-TK, MXene-TK-DOX, and MXene-TK-DOX@PDA nanoparticles were estimated as −42.8, −51.8, and −47.8 mV, respectively. This polarity reversion suggests that TK successfully reacted with APTES-functionalized MXene because of the presence of a carboxyl group in the TK linker. Therefore, the MXene-TK-DOX and MXene-TK-DOX@PDA nanoparticles showed negative charges, which were ascribed to the phenolic hydroxyl and catechol groups, respectively. These results further indicate the successful synthesis of MXene-TK-DOX@PDA nanoparticles.

The FTIR spectra (Figure 2d and Appendix A) of the pristine MXene, MXene-NH_2_, MXene-TK, MXene-TK-DOX, and MXene-TK-DOX@PDA samples are presented to clarify the organic compounds (APTES, TK, DOX, and polydopamine) attached to the MXene surface. The vibrational peaks of the OH group of pristine MXene (Appendix A) were observed at 3434 cm^−1^, and the peak at 540 cm^−1^ corresponded to the Ti–O–Ti stretching vibration [51]. The characteristic peaks at 2913 and 3415 cm^−1^ for MXene-NH_2_ were assigned to the symmetric stretching vibrations of -CH_2_ and the stretching vibrations of N–H (–NH_2_), respectively. Furthermore, the characteristic Si–O peak was observed at 765 cm^−1^. The results indicate the immobilization of APTES onto the surface of the MXene (Ti_3_C_2_) nanosheets. The characteristic absorption peaks of MXene-TK and MXene-TK-DOX are shown in Appendix A. For MXene-TK, new peaks at 1723 and 1637 cm^−1^ corresponded to the ester C=O and amide groups, respectively, suggesting that the ROS-cleavable TK-based linker was grafted onto the MXene surface. Meanwhile, the peaks at 3421, 2917/2850, 1090, and 1623 cm^−1^ observed for the MXene-TK-DOX nanomaterials were attributed to the stretching vibration of N–H, C–H, C–O–C, and the ring stretching vibration of C=C, respectively, which further indicates the introduction of DOX on to the MXene surface. After coating with pH-responsive polydopamine, the peaks observed at 1290 and 1615 cm^−1^ corresponded to the C–O (phenolic) and C=C peaks, respectively, as shown in Appendix A. Therefore, FTIR analysis demonstrates the successful preparation of the MXene-TK-DOX@PDA nanoparticles.

The structural characteristics of the Ti_3_AlC_2_ and MXene + TBAOH nanosheets were further investigated using Raman spectroscopy (Figure 2e,f). The characteristic peaks of Ti_3_AlC_2_ were detected at 269, 416, and 611 cm^−1^ due to the shear and longitudinal oscillations of titanium and aluminum atoms, which is consistent with previous reports [52]. Notably, the characteristic peak of the vibrations of aluminum atoms observed at 269 cm^−1^ for the Ti_3_AlC_2_ powder disappeared after HF etching, which indicates that the aluminum layer was successfully exfoliated. Furthermore, the scattering peak of MXene + TBAOH was observed at 150 cm^−1^, which was associated with the existence of oxidized Ti_3_C_2_ after etching. The presence of graphitic carbon and amorphous carbon in the MXene + TBAOH nanosheets is indicated by the overlapping peaks at 1370 and 1563 cm^−1^, which corresponded to the D and G bands, respectively [53]. These results demonstrate that Ti_3_AlC_2_ was successfully etched and delaminated with TBAOH.

Additionally, the hydrodynamic diameter distributions of the MXene, MXene-TK, MXene-TK-DOX, and MXene-TK-DOX@PDA nanoparticles were determined via dynamic light scattering (Figure 2g) as 142, 153, 165, and 176 nm, respectively. Notably, the hydrodynamic diameter increased after the reaction with organic molecules, implying that APTES, TK, DOX, and polydopamine were introduced onto the surface of MXene. The size of the MXene-TK-DOX@PDA nanoparticles has been reported to be approximately 176 nm, which is favorable for accumulating on tumor sites due to the enhanced permeability and retention (EPR) effect [54].

The lateral size and thickness of the intercalated MXene nanosheets were determined using AFM, as shown in Figure 2h. A highly dispersed two-dimensional nanostructured MXene nanosheet was also observed. The lateral size and thickness of MXene nanoflakes were approximately 200 nm and 1.9 nm, respectively, as depicted in Figure 2j. Furthermore, the Tyndall effect of colloidal MXene is presented in Figure 2i, which exhibited a bright-light path, indicating its excellent dispersity and good affinity in aqueous colloidal solutions.

### 3.3. Morphological Analysis of MXene-Based Nanomaterials

Two-dimensional-structured MXenes with a large number of anchor sites and a broad specific surface area are excellent delivery systems in biological applications [55]. Figure 3 shows the morphologies of the Ti_3_AlC_2_, MXene (Ti_3_C_2_), MXene + TBAOH, MXene -NH_2_, MXene-TK, and MXene-TK-DOX@PDA nanoparticles. Ti_3_AlC_2_ exhibited a compact layered structure with a micron-sized in Figure 3a. After etching with 40% HF, an accordion-like structure was obtained, possibly because of the high reactivity between the Al atomic layer of Ti_3_AlC_2_ and F ions as well as the large amount of escaped H_2_ due to the exothermic nature of the HF reaction with aluminum (Figure 3b) [56]. After TBAOH intercalation, ultrathin flakes were observed (Figure 3c) because of the incorporation of the TBAOH intercalator into the interlayer of pristine MXene, which is consistent with a previous report [57]. After functionalization with APTES, an irregular distribution of nanosized flakes (MXene-NH_2_) was observed, which is probably due to the electrostatic interaction between the amine groups of APTES (Figure 3d). Additionally, the EDX mapping of MXene-NH_2_ showed two new elements (highlighted in red; Appendix A). The Si and N contents in MXene-NH_2_ were estimated to be 0.42 and 6.54 wt.%, respectively, indicating the successful coupling of APTES with the MXene surface. Moreover, the SEM image of MXene-TK (Figure 3e) revealed that the introduction of the ROS-cleavable TK had no significant influence on the morphology compared with the case of the MXene-NH_2_ nanosheets. Additionally, the S content in MXene-TK was 0.09 wt.%, which indicates that the ROS-cleavable linker was successfully introduced, as described in Appendix A. Furthermore, MXene-TK-DOX@PDA (Figure 3f) exhibited a nanoparticle morphology, which was ascribed to the pH-responsive polydopamine coating on the surface of MXene-TK-DOX via stirring. Additionally, the interaction of the polydopamine functional groups could efficiently prevent their agglomeration.

Furthermore, EDX mapping analysis of the MXene-TK-DOX@PDA nanoparticles proved that APTES, TK, DOX, and polydopamine were successfully grafted onto the surface of the MXene nanosheets. As shown in Figure 4, the N and Si contents in the MXene-TK-DOX@PDA nanoparticles were 8.35 and 0.4 wt.%, respectively, which indicates the successful conjugation of APTES on to the MXene surface. Moreover, a S content of 0.32 wt.% from the TK component was detected. Notably, the N content changed from 6.54 to 8.35 wt.%, which could be due to the presence of DOX and the pH-responsive polydopamine. These results demonstrate the successful synthesis of the MXene-TK-DOX@PDA nanoparticles.

### 3.4. In Vitro Stability of MXene-Based Nanoparticles

The stability of the drug delivery system is critical for efficient drug delivery in vivo [58]. Therefore, the stability of MXene-TK-DOX and MXene-TK-DOX@PDA was evaluated in a mimetic physiological environment (PBS 7.4) for seven days. As shown in Figure 5a, MXene-TK-DOX showed good dispersion within 1 d, which may be attributed to the nanosized particles and hydrophilic terminated groups (-O, -OH, and/or-F). However, significant agglomeration and precipitation of MXene-TK-DOX were observed after 24 h of standing, and an increase in the precipitation rate was noted with time, probably due to the oxidation process on the surface, which transformed the hydrophilic groups to hydrophobic groups [59]. Compared with the above-mentioned product, the MXene-TK-DOX@PDA nanoparticles (Figure 5b) exhibited superior colloidal stability and dispersibility in a mimetic physiological environment after 7 d of retention due to the mussel-inspired biomimetic material polydopamine with hydrophilic amino and phenolic hydroxyl groups, which is plausibly favorable for prolonging the blood circulation time in vivo. Additionally, the Tyndall effect of MXene-TK-DOX@PDA nanoparticles (in lateral and top views) was also monitored after 7 d of standing (Figure 5c,d, respectively). The results demonstrate the good colloidal stability of the product in a physiological solution.

### 3.5. XPS Analysis

The chemical composition, electronic states, and bond formation of the MXene nanosheets and MXene-TK-DOX@PDA nanoparticles were evaluated using XPS. The survey scan spectra of MXene-TK-DOX@PDA and MXene + TBAOH (Figure 6) revealed that the binding energies of It, C, O, Si, N, F, and S were 456, 285, 531, 685, 102, 401, and 163 eV, respectively. The high-resolution XPS spectra of Ti 2p, C 1s, O 1s, Si 2p, N 1s, F 1s, and S 2p of the MXene-TK-DOX@PDA nanoparticles are shown in Figure 7a-g, respectively. The Ti 2p XPS spectrum of the MXene-TK-DOX@PDA nanoparticles was fitted to reveal Ti–C, Ti(II), and Ti(III) components at binding energies of 454.5, 456.2, and 455.3, respectively (Figure 7a). The Ti 2p_3/2_ XPS spectrum showed two peaks at 458.6 and 460 eV, corresponding to TiO_2_ and TiO_2-x_F_x_, respectively [60]. The Ti 2p XPS spectrum (Appendix A) of MXene + TBAOH showed three peaks at 454.4, 456.2, and 455.2 eV, corresponding to Ti–C, Ti(II), and Ti(III), respectively. Compared to MXene-TK-DOX@PDA, the binding energies of the Ti 2p XPS spectrum of MXene + TBAOH showed a decreasing trend, which may indicate the changes in the chemical composition of the Ti atoms. The C 1s XPS spectrum of the MXene-TK-DOX@PDA nanoparticles exhibited peaks at 281.2, 284.2, 285, and 286.6 eV, corresponding to the C–Ti, C–C/C–H, C–O, and O–C=O bonds [61], respectively, as depicted in Figure 7b. The MXene-TK-DOX@PDA O 1s XPS spectrum (Figure 7c) showed three peaks with binding energies of 532.3, 531.1, and 529.8 eV, which were associated with C–Ti–(OH)_x_, CTiO_x_, and O–Ti bonds, respectively. Furthermore, the O 1s XPS spectrum of MXene + TBAOH (Appendix A) was deconvoluted into three peaks at 532.5, 531.1, and 529.9 eV, which corresponded to the C–Ti–(OH)_x_, C−TiO_x_, and O−Ti bonds, respectively. After APTES grafting, the high-resolution Si 2p XPS spectrum of the MXene-TK-DOX@PDA nanoparticles showed peaks at 102 and 101.1 eV, corresponding to Si–O–Si and Si-C, respectively (Figure 7d) [62]. Therefore, the results indicate that APTES was successfully conjugated to the MXene surface. In the N 1s XPS spectrum of the MXene-TK-DOX@PDA nanoparticles (Figure 7e), four peaks were observed at 400, 399.3, 401, and 401.8 eV, respectively, which were assigned to the N–C bond and the primary, secondary, and tertiary amines, respectively. Significantly, the N 1s XPS spectrum of MXene + TBAOH showed a characteristic peak corresponding to TBAOH (Appendix A). The F 1s XPS spectrum of the MXene-TK-DOX@PDA nanoparticles (Figure 6g) exhibited two deconvoluted peaks at 685.1 and 684.4 eV, which were attributed to TiO_2-x_F_x_ and the C–Ti–F bond, respectively. Furthermore, the F 1s XPS spectrum of MXene + TBAOH is presented in Appendix A. For the S 2p spectrum of the MXene-TK-DOX@PDA nanoparticles, two peaks corresponding to the S 2p_3/2_ and S 2p _5/2_ orbitals were observed (Figure 7g) at 164 and 166 eV, respectively [63]. Figure 7h shows the high-resolution C 1s XPS spectrum of the MXene + TBAOH nanosheets, which was deconvoluted into four peaks with binding energies centered at 288.6, 285, 284.2, and 281.2 eV, corresponding to O–C=O, C–O, C–C/C–H, and C–Ti, respectively. Therefore, the XPS spectra of Ti 2p, O 1s, Si 2p, N 1s, and S 2p revealed the successful preparation of the MXene-TK-DOX@PDA nanoparticles.

### 3.6. In Vitro Photothermal Performance of MXene-Based Nanoparticles

A UV–Vis spectroscopy was employed to characterize DOX loading (Figure 8a). Compared with the MXene-TK nanosheets, the MXene-TK-DOX nanoparticles showed a stronger characteristic absorption peak at 450–550 nm, which revealed that DOX was successfully grafted onto the surface of MXene. Notably, the characteristic DOX absorption peak of the MXene-TK-DOX nanoparticles exhibited a red shift (from 495 to 505 nm) compared with that of pristine DOX because of the interaction between the DOX and MXene nanosheets [64], which was similar to the observations for DOX conjugated to other two-dimensional materials (such as graphene and black phosphorus) and is consistent with a previous report [65].

The in vitro photothermal conversion efficacy of the MXene-TK-DOX@PDA nanoparticles was evaluated by monitoring the temperature variations in the sample’s aqueous solution after exposure to 808 nm-laser irradiations at a power density of 2.0 W cm^−2^ (Figure 8b). The temperature difference (ΔT) of the MXene-TK-DOX@PDA nanoparticles in an aqueous solution reached 45 °C after 10 min of irradiation at a concentration of 300 µg/mL. However, the actual temperature at the tumor site may reach 65 °C (as hypothesized for an ambient temperature of 20 °C). Hao et al. [66] measured the photothermal conversion efficiency of Prussian blue analogues at (PBAs) around 38 °C after exposure to 808 nm-laser irradiations for 10 min at aa concentration of 100 μg/mL. Jin et al. [67] showed the photothermal conversion properties of monolayer Bi-anchored manganese boride nanosheets (MBBN), a two-dimensional material of metal boride, at around 48 °C after exposure to 808 nm-laser irradiations for 10 min at a concentration of 100 μg/mL. These previous works clearly show that MXene-based nanomaterials exhibit excellent photothermal conversion performance. Notably, to realize an effective ablation of tumor tissue, temperatures exceeding 50 °C are required to overcome the thermal resistance caused by heat-shock proteins [68,69]. The temperature of the pure water solution did not significantly change after exposure to 808 nm-laser irradiations for 10 min, as depicted in Figure 8b. However, the temperature of the MXene-TK-DOX@PDA nanoparticles was dramatically enhanced, which demonstrated that the NIR light can be efficiently and rapidly converted to thermal energy in the presence of MXene-TK-DOX@PDA nanoparticles. Additionally, the photothermal conversion properties of pristine MXene nanosheets were evaluated (Appendix A) at various concentrations (50, 100, 200, and 300 µg/mL). At an MXene nanosheet concentration of 200 µg/mL, the ΔT of the solution increased by only 11 °C after 5 min of irradiation with an 808 nm-laser (2 W/cm^2^). However, the ΔT of the MXene-TK-DOX@PDA nanoparticles reached 39 °C in 5 min at the same concentration (200 µg/mL). This may be attributed to the polydopamine biomaterials, which possess salient photothermal conversion properties in the presence of laser illumination [70]. Notably, MXene does not show a high photothermal conversion efficiency at a concentration of 300 ug/mL, which may be because of the sedimentation of MXene. However, the MXene-TK-DOX@PDA nanoparticles exhibited high photothermal conversion efficiency at the same concentration (300 ug/mL), as shown in Figure 8, which further demonstrates that PDA can improve the biostability of the drug delivery system, which is consistent with the previous results.

The photothermal conversion efficiency of the MXene-TK-DOX@PDA nanoparticles was further explored at different power densities (increasing from 0.5 to 2.0 W cm^−2^), as shown in Figure 8c, at a fixed concentration of 300 ug/mL. The ΔT of the MXene-TK-DOX@PDA nanoparticles in an aqueous solution reached 42 °C as the laser power density increased to 2.0 W cm^−2^, which is sufficient to ablate the cancer cells. To estimate photothermal stability, five heating/cooling cycles were performed for the MXene-TK-DOX@PDA nanoparticles, as depicted in Figure 8d. No apparent temperature decrease was observed after five cycles, demonstrating that the MXene-TK-DOX@PDA nanoparticles have high photothermal stability and can be used as durable photothermal agents for photothermal therapy (PTT). The absorbance spectra of the MXene nanosheets obtained at various concentrations (2, 4, 8, 15, and 30 mg/L) exhibited a distinct absorption peak (Figure 8e) at the first biological window (~808 nm), which is favorable for their application as photothermal agents for PTT [71]. Additionally, the extinction coefficient of the MXene nanosheet was measured based on a linear fit of the MXene concentration (2, 4, 8, 15, and 30 mg/L) and A/L (A denotes the absorbance of MXene at 808 nm, and L denotes path-length = 1 cm) ratio at 808 nm (Figure 8f). Based on the Lambert–Beer law, A/L = αC (where C denotes the concentration of MXene; α denotes the extinction coefficient), the extinction coefficient of MXene was estimated to be 23.31 Lg^−1^cm^−1^, which was higher than that of other two-dimensional materials. Liu et al. calculated the extinction coefficient of the Ta_4_C_3_-IONP-SPs nanoplatform to be approximately 4 Lg^−1^cm^−1^ [72]. Robinson et al. reported that the single-layered nano-reduced graphene oxide (rGO) sheets had an ∼3.6 Lg^−1^cm^−1^ in the extinction coefficient [73]. Lin et al. claimed that the extinction coefficient at 808 nm was measured to be 4.06 Lg^−1^cm^−1^ of the Ta_4_C_3_ nanosheets [74]. Based on these previous works, it can be said that MXene nanosheets show a distinctive advantage as promising photothermal agents for cancer therapy in clinical applications. Notably, the photothermal conversion efficiency of an agent depends on two main parameters: extinction coefficient (ε) and photothermal conversion efficiency (η). The extinction coefficient and photothermal-conversion efficiency reflect the light absorption ability and conversion performance for converting light energy into thermal energy, respectively [75].

Infrared thermal images of the MXene-TK-DOX@PDA nanoparticles were acquired using a thermal imaging camera after exposure to 808 nm-laser irradiations (2.0 W cm^−2^) at a concentration of 1 mg/mL. As shown in Figure 9, the temperature of the MXene-TK-DOX@PDA nanoparticles showed an increasing trend from 27.6 °C to 52.9 °C with an increasing laser irradiation time. Notably, the temperature could reach up to 52.9 °C after 10 min of irradiation, which is sufficiently high to ablate the tumor. The photothermal conversion properties of the MXene-TK-DOX@PDA nanoparticles are based on two main parameters: the photothermal behavior of MXene nanosheets and that of polydopamine. MXene has a localized surface plasma resonance (LSPR) effect [76], while polydopamine indicates the existence of donor–acceptor molecular pair structures, which could decrease the energy bandgap and increase the electron delocalization for polydopamine, thereby enhancing light absorption across a broad spectrum [77]. Furthermore, the infrared thermal images of the MXene nanosheets were evaluated at various concentrations (50, 100, 200, and 300 µg/mL), as depicted in Appendix A. At a low concentration (50 µg/mL), MXene exhibited a weak photothermal conversion property (only a 10.6 °C temperature difference) in 10 min. However, the temperature of MXene reached as high as 54.3 °C as the concentration increased to 200 µg/mL. In addition, a 27.1 °C temperature difference revealed the superior photothermal conversion efficiency of MXene.

### 3.7. In Vitro Antibacterial Activity

Various pathogenic and multidrug-resistant strains of bacteria are generally considered the two main factors that have caused substantial morbidity and mortality in clinical settings, posing a serious threat to public health [78]. Therefore, there is an urgent need to design highly effective antimicrobial nanomaterials to prevent the increasing prevalence of bacterial infections. Hence, the antibacterial effects of the MXene, MXene-TK-DOX, and MXene-TK-DOX@PDA nanoparticles with a concentration of 6.085 mg/mL were investigated using both Gram-negative (*E. coli*) and Gram-positive (*B. subtilis*) bacteria. Initially, the microbe was activated in the Luria–Bertani medium and diluted 200-fold. The supernatant was then dispersed in the sample solution, and the antibacterial ability of the products was determined using the colony counting method. A large number of *E. coli* and *B. subtilis* colonies were observed in the control group after 5 h of incubation (Figure 10); this revealed that the *E. coli* and *B. subtilis* bacteria were well-activated, possessed high bacterial viability, and indicated negligible antimicrobial effects. In contrast, the MXene and MXene-TK-DOX treated groups exhibited high antibacterial efficiency and completely killed (~100% reduction) both bacteria (*E. coli* and *B. subtilis*) after 5 h of incubation. This is attributable to both nanosheets with sharp edges, which may be readily entrapped into the surface of the bacteria and lead to membrane destruction. The results further demonstrate that the antibacterial properties of MXene are not significantly influenced by modification with thioketal and doxorubicin. Photographs of the treated plates for *E. coli* and *B. subtilis* colonies exposed to MXene-TK-DOX@PDA nanoparticles revealed that they also possessed high growth inhibition abilities for both bacteria after 5 h of incubation. This result can be explained on the basis of the antimicrobial properties of polydopamine. Notably, the antibacterial property of PDA depends on two main mechanisms. One of the mechanisms is that PDA can rapidly entrap bacteria and interact with their secreted proteins, following which the metabolism of the bacteria is impeded [79]. Another mechanism is that PDA can destroy the matrix of biofilms in the presence of light; this can be attributed to the photothermal antibacterial ability of polydopamine [80]. These findings indicate that MXene-based nanomaterials have a superior inhibitory effect on both bacteria (*E. coli* and *B. subtilis*).

### 3.8. In Vitro Drug Release Profile

The loading content and encapsulation efficiency of DOX from the MXene-TK-DOX@PDA nanoplatforms were calculated as 13.4% and 27%, respectively, using Equations (1) and (2) with the aid of the calibration curve, as shown in Appendix A. The DOX release percentage of MXene-TK-DOX@PDA nanoparticles in various media was measured using Equation (3) and is shown in Figure 11. A drug release percentage of 5.04% was observed in a mimetic physiological environment (pH of 7.4, PBS) after 48 h; this revealed that the MXene-TK-DOX@PDA nanoparticles could efficiently avoid the unexpected release of DOX during systemic circulation. Furthermore, the percentage of DOX released from the MXene-TK-DOX@PDA nanoparticles remained low in a weakly acidic environment (pH 5.5). This may be because DOX is chemically bonded to the MXene surface [81], effectively avoiding premature release of the drug. It is well-known that thioketals (TK) exhibit ROS-responsive properties. Therefore, the ROS-rupturable behavior of the MXene-TK-DOX@PDA nanoparticles was investigated in H_2_O_2_ solutions with different pH (5.5 and 7.4). At pH 7.4, the cumulative DOX release percentage from the MXene-TK-DOX@PDA nanoparticles was 43.07% in the presence of H_2_O_2_ for 48 h, which indicated the cleavage of thioketal and the release of DOX. We not only investigated the drug release rate under physiological conditions (pH 7.4), but also studied the DOX release properties in the tumor microenvironment (pH 5.5). As expected, the MXene-TK-DOX@PDA nanoparticles showed a higher drug release percentage (61.8%) in H_2_O_2_ solution at pH 5.5 after 48 h of incubation. This is attributable to the thioketal easily cleaving to the H_2_O_2_ environment [23] and the possibly higher solubility of DOX in a weak acid solution compared with under neutral conditions [39]. The results of the release performance of the MXene-TK-DOX@PDA nanoparticles revealed that the nanosized vector could control the release of a drug and exhibited good ROS-responsive and pH-responsive behavior.

### 3.9. Drug Release Kinetics

The release mechanism of DOX from the MXene−TK−DOX@PDA nanoparticles was studied using the Korsmeyer–Peppas and Higuchi models. The parameters of the Korsmeyer–Peppas model release mechanism is listed in Table 1. The Korsmeyer–Peppas empirical equation is as follows [82]:Korsmeyer–Peppas: M_t_/M_∞_ = k_p_t^n^
(4)
where M_t_/M_∞_ is the percentage of drug released at time t; k_p_ is the rate constant; and n is the release index.

Compared with the Higuchi model, which is described in the Appendix A, the release kinetics of DOX from the MXene-TK-DOX@PDA nanoparticles fit the Korsmeyer–Peppas model better because of a higher correlation coefficient (R^2^). In addition, the release exponent (n) values were 0.258, 0.184, 0.481, and 0.416 in the presence of pH 5.5, 7.4, pH 7.4 + H_2_O_2_, and 5.5 + H_2_O_2_, respectively. The release index results indicate that the release kinetics of the MXene-TK-DOX@PDA nanoparticles mainly followed the Fickian diffusion mechanism because the n value was less than 0.45 [83]. These results suggest that Fickian diffusion plays a major role in the release process. The DOX release index was explored using the Higuchi model [84], and the results are shown in Appendix A. Moreover, smaller R² values were observed in the Higuchi model. The Higuchi model was found to be unsuitable for the release mechanism of the MXene-TK-DOX@PDA nanoparticles.

## 4. Conclusions

In this study, novel MXene-based nanoparticles with a lateral size and thickness of approximately 200 nm and 1.9 nm, respectively, were proposed and designed for chemotherapeutic drug targeting delivery along with excellent antibacterial activity against Gram (−) *E. coli* and Gram (+) *B. subtilis* in vitro. To meet the requirements of biomedical applications, hydrofluoric acid etching and TBAOH intercalation were used to obtain well-dispersed MXenes with small lateral dimensions. Infrared thermal images indicate that the MXene-TK-DOX@PDA nanoparticles possess salient photothermal conversion properties and high photothermal stability. This is attributable to the high extinction coefficient of MXene (23.31 L/g/cm), which could act as a long-lasting photothermal agent for photothermal therapy. Furthermore, the MXene-TK-DOX@PDA nanoparticles exhibited sensitive ROS- and pH-response behaviors in the presence of H_2_O_2_ at pH 5.5. Notably, the MXene-TK-DOX@PDA nanoparticles controlled the locoregional release of DOX and effectively reduced the premature release of drugs. Furthermore, the MXene-TK-DOX@PDA nanoplatform exhibited a significant inhibition of Gram-negative *E. coli* and Gram-positive *B. subtilis* within 5 h in vitro. We expect that the MXene-TK-DOX@PDA nanoparticles will potentially afford a promising strategy for drug delivery in the biomedical field and enrich the family of antibacterial materials.

## Figures and Tables

**Figure 1 nanomaterials-12-04392-f001:**
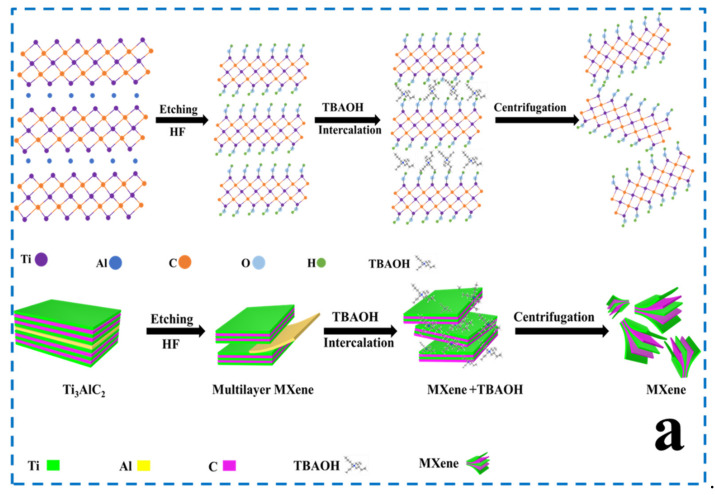
(**a**) Schematic representation of the MXene synthetic process including etching, intercalation, and centrifugation. (**b**) Illustration of the preparation process of MXene−TK−DOX@PDA nanoparticles by reaction with APTES, thioketal, doxorubicin, and coating with PDA, respectively. (**c**,**d**) TEM images of the pristine MXene nanosheets before and after delamination by TBAOH. (**e**) High resolution TEM image of the MXene nanosheets after HF etching (d = 1.06 nm). (**f**) High resolution TEM image of the MXene + TBAOH nanoparticles (d = 1.54 nm). (**g**) TEM image of the MXene−TK−DOX@PDA nanoparticles. (**h**) HRTEM image of MXene and selected area electron diffraction (SAED) (inset). (**i**) Illustration of the process of MXene synthesis and the doxorubicin release from MXene−TK−DOX@PDA nanoparticles.

**Figure 2 nanomaterials-12-04392-f002:**
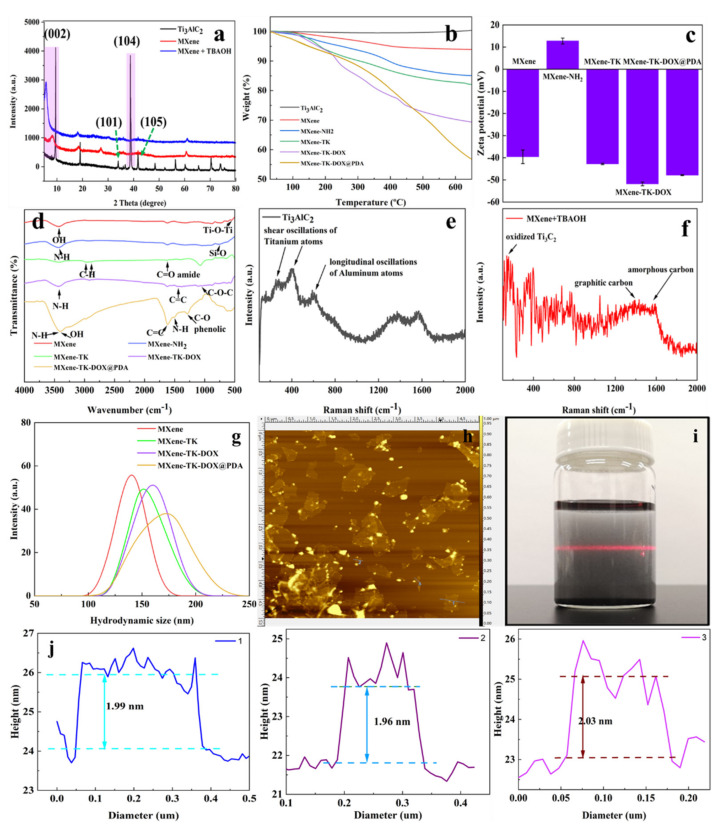
(**a**) XRD patterns of the Ti_3_AlC_2_, MXene, and MXene + TBAOH nanoparticles. (**b**) TGA curves of the Ti_3_AlC_2_, MXene, MXene−NH_2_, MXene−TK, MXene−TK−DOX, and MXene−TK−DOX @PDA nanoparticles. (**c**) Zeta potentials of the MXene, MXene-NH_2_, MXene−TK, MXene−TK−DOX, and MXene−TK−DOX@PDA nanoparticles. (**d**) FTIR spectra of the MXene, MXene−NH_2_, MXene−TK, MXene−TK−DOX, and MXene−TK−DOX@PDA nanoparticles. (**e**,**f**) Raman spectra of Ti_3_AlC_2_ and MXene + TBAOH. (**g**) Hydrodynamic diameters othe f MXene, MXene−TK, MXene−TK−DOX, and MXene−TK−DOX@PDA nanoparticles. (**h**,**j**) Thickness and lateral size of the MXene nanosheets after delamination as measured using AFM. (**i**) Optical image showing the Tyndall effect of MXene dispersed in an aqueous colloidal solution.

**Figure 3 nanomaterials-12-04392-f003:**
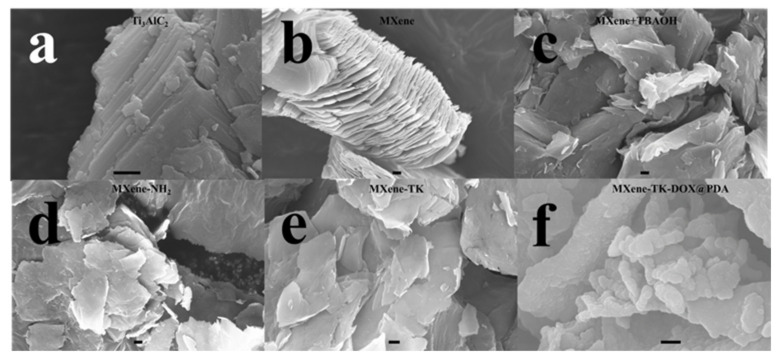
FE-SEM images of the (**a**) Ti_3_AlC_2_, (**b**) MXene (Ti_3_C_2_), (**c**) MXene + TBAOH, (**d**) MXene−NH_2_, (**e**) MXene−TK, and (**f**) MXene−TK−DOX@PDA nanoparticles. Scale bars: 300 nm.

**Figure 4 nanomaterials-12-04392-f004:**
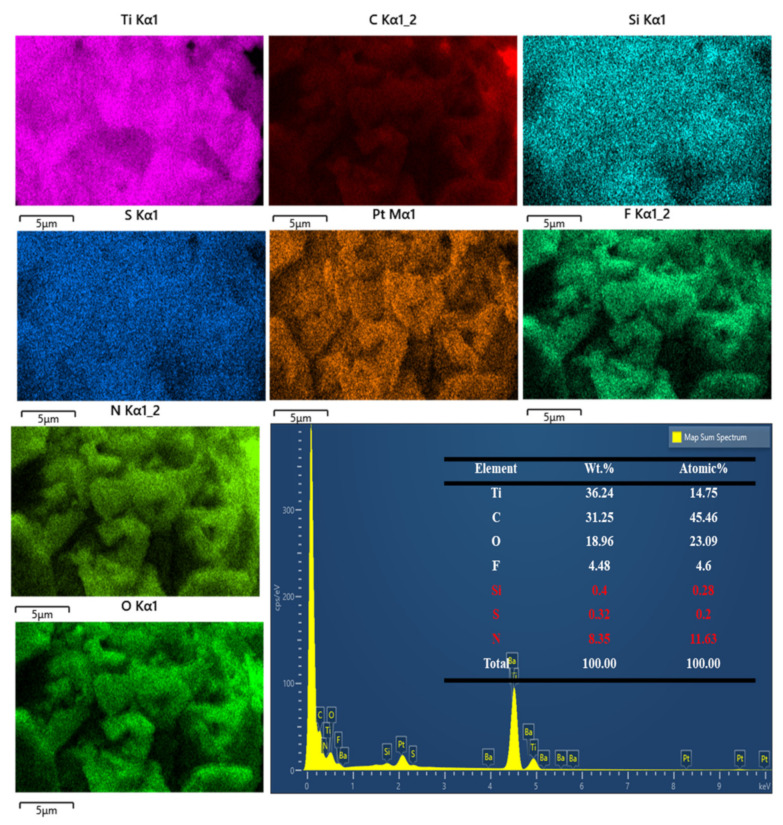
EDX mapping of the MXene−TK−DOX@PDA nanoparticles.

**Figure 5 nanomaterials-12-04392-f005:**
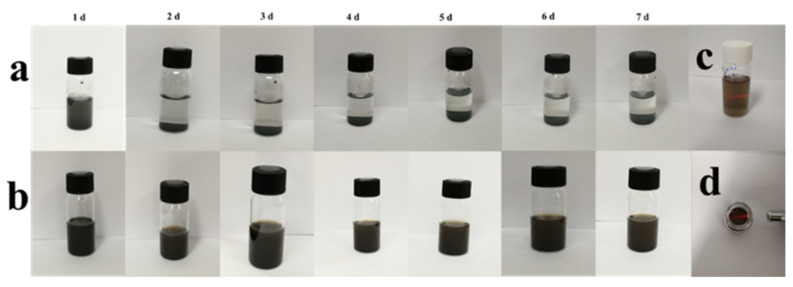
Stability of the MXene−based nanomaterials in a mimetic physiological environment. (**a**) MXene−TK−DOX and (**b**) MXene−TK−DOX@PDA nanoparticles. (**c**,**d**) Side and top views showing the Tyndall effect of the MXene−TK−DOX@PDA nanoparticles after an incubation period of 7 d, respectively.

**Figure 6 nanomaterials-12-04392-f006:**
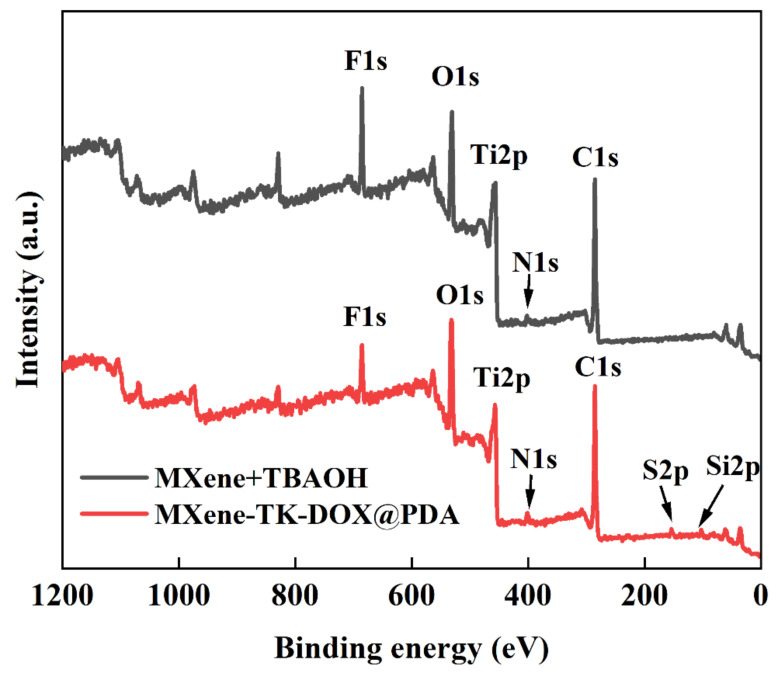
Survey scan spectra of the MXene + TBAOH and MXene−TK−DOX@PDA nanoparticles.

**Figure 7 nanomaterials-12-04392-f007:**
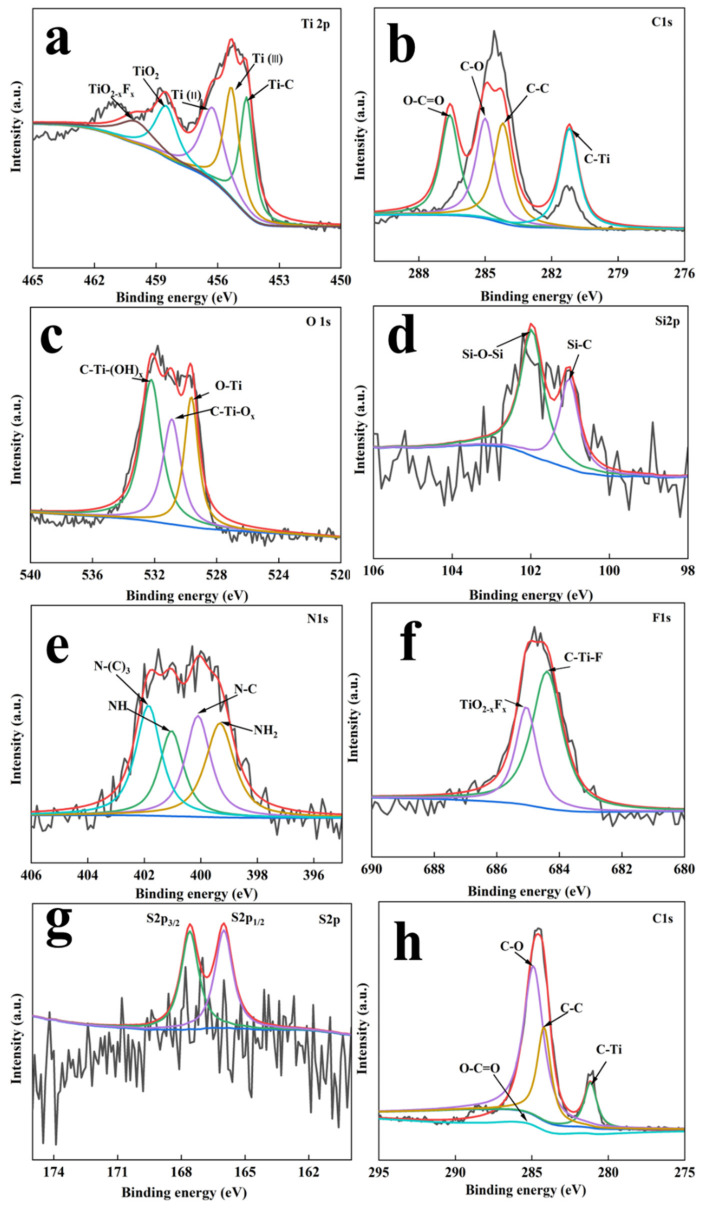
XPS spectra of the MXene−TK−DOX@PDA nanoparticles. (**a**–**g**) Ti 2p, C 1s, O 1s, Si2p, N1s, F 1s, and S 2p spectra of MXene−TK−DOX@PDA. (**h**) C 1s XPS spectrum for MXene + TBAOH.

**Figure 8 nanomaterials-12-04392-f008:**
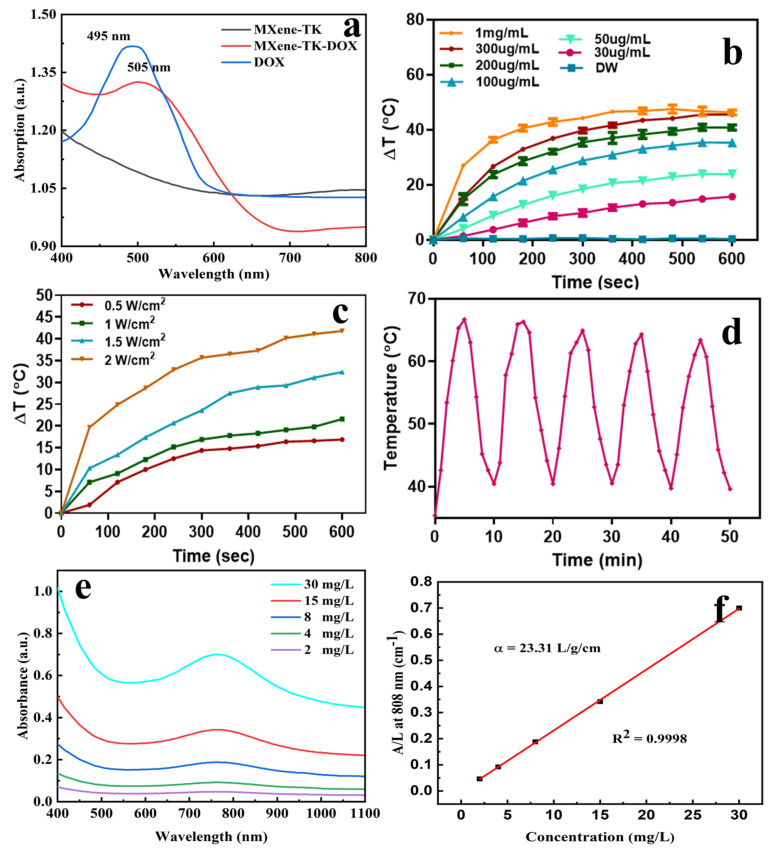
In vitro photothermal-property characterization of the MXene−based nanoparticles under various conditions. (**a**) UV−Vis absorption spectra of the pure doxorubicin, MXene−TK, and MXene−TK−DOX nanoparticles. (**b**) Photothermal conversion profile of the MXene−TK−DOX@PDA nanoparticles in an aqueous solution after 808 nm laser irradiation (2 W/cm^2^) at elevated concentrations (0, 30, 50, 100, 200, 300 ug/mL, and 1 mg/mL). (**c**) Photothermal property curve of the MXene−TK−DOX@PDA nanoparticles in an aqueous solution under 808 nm laser irradiation at 300 ug/mL in the presence of different laser densities (0.5, 1, 1.5, and 2 W/cm^2^). (**d**) Recycling heating curves of the MXene−TK−DOX@PDA nanoparticles in an aqueous solution after 808 nm laser irradiation at 2 W cm^−2^ for five laser on/off cycles. (**e**) UV−Vis−NIR absorption spectra of the MXene nanosheets at diverse concentrations (2, 4, 8, 15, and 30 mg/L). (**f**) Extinction coefficient of the MXene nanosheets.

**Figure 9 nanomaterials-12-04392-f009:**
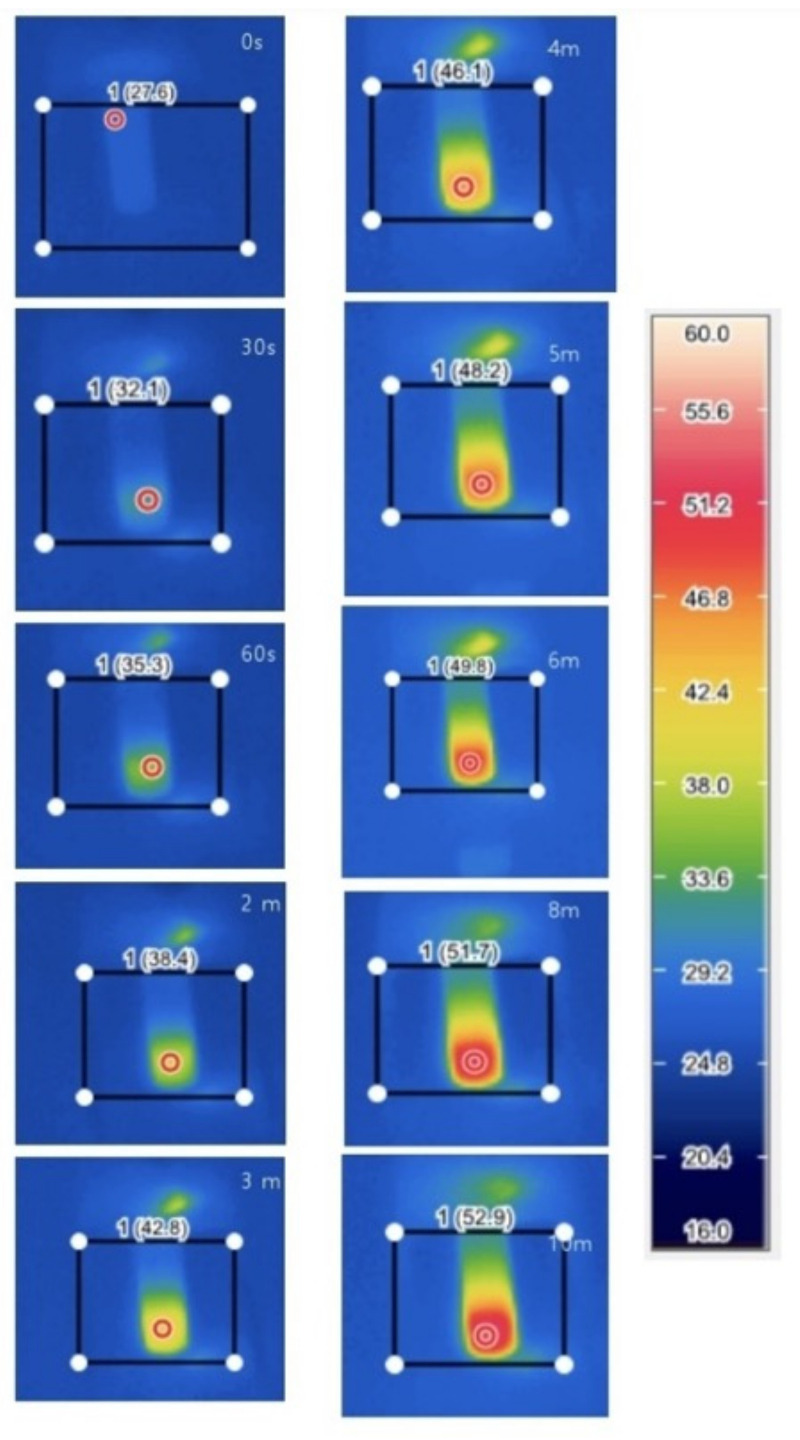
Infrared thermal images of the MXene−TK−DOX@PDA nanoparticles at a concentration of 1 mg/mL under laser (2 W/cm^2^ at 808 nm) illumination for 0, 30, 60 s, 2, 3, 4, 6, 8, and 10 min, respectively.

**Figure 10 nanomaterials-12-04392-f010:**
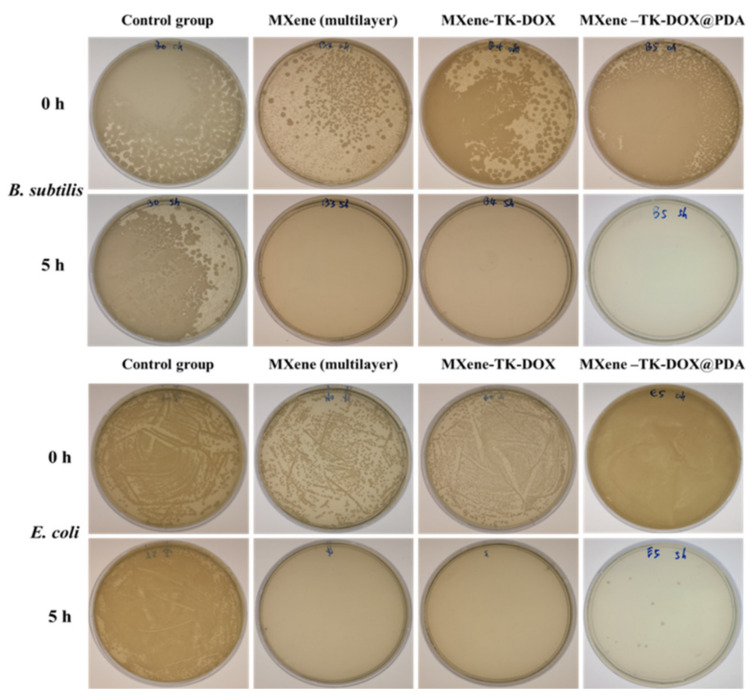
Antibacterial activities in aqueous solutions after 5 h of incubation: bacterial suspensions in PBS solution (0.01 mM) acted as the control group. A photograph of the test plates for *E. coli* and *B. subtilis* bacteria exposed to the MXene, MXene−TK−DOX, and MXene−TK−DOX@PDA nanoparticles at a concentration of 6.085 mg/mL, respectively. All analyses of the results were performed in triplicate.

**Figure 11 nanomaterials-12-04392-f011:**
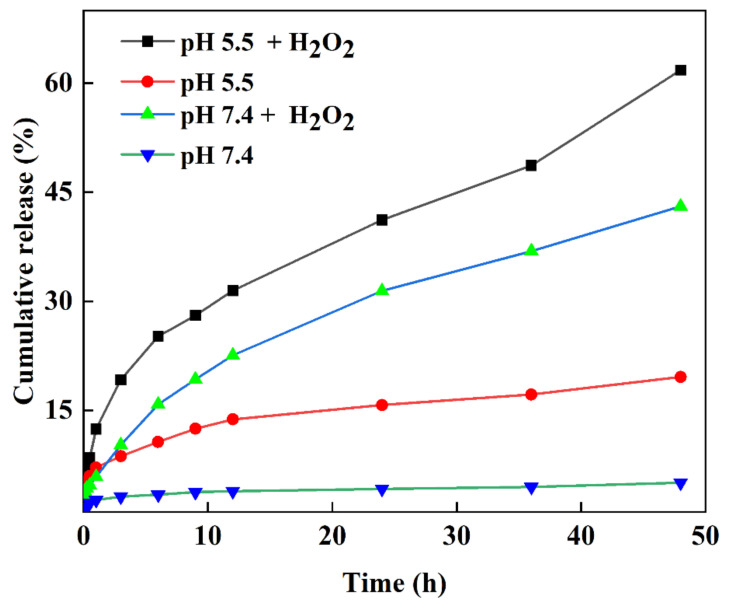
The drug release curves from the MXene−TK−DOX@PDA nanoparticles in different media.

**Table 1 nanomaterials-12-04392-t001:** In vitro doxorubicin release kinetics data based on the Korsmeyer–Peppas model.

Sample Code	Release Medium	Korsmeyer–Peppas		
		n	k_p_*	R^2^
	pH 7.4	0.184	2.414	0.944
	pH 7.4 + H_2_O_2_	0.481	6.686	0.991
MXene-TK-DOX@PDA	pH 5.5	0.258	7.005	0.960
	pH 5.5 + H_2_O_2_	0.416	11.56	0.985

k_p_* represents the release rate constant; n represents the release exponent; and R^2^ represents the correlation coefficient.

## Data Availability

Not applicable.

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
