# Peer review of "ROS- and pH-Responsive Polydopamine Functionalized Ti3C2Tx MXene-Based Nanoparticles as Drug Delivery Nanocarriers with High Antibacterial Activity"

_nanomaterials, 2022, doi:10.3390/nano12244392_

Round 1

Reviewer 1 Report

The manuscript entitled " ROS- and pH-Responsive Polydopamine Functionalized Ti3C2Tx MXene-Based Nanoparticles as Drug Delivery Nanocarriers with High Antibacterial Activity" authored by Wei-Jin Zhang and co-authors reports on functionalized MXene-derived nanoparticles as nanovectors for antibacterial drug delivery. They used several microscopic and spectroscopic  techniques to investigate chemical properties and to characterise their morphology\size\charges and so on. The paper is well structured and fulfils totally the aim and interest of Nanomaterials broad audience. I would reccomend just minor changes\revisions in order to reach standard for publication In nanomaterials and to improve the quality of manuscript. Below there is a (short) list of suggestions\reccomendations\issues that authors should follow \address before re-submitting it:

1) I would suggest a comprehensive literature update by adding few research articles\reviews comparing them with competitive nanocarriers in the market\scientific literature.

2) A moderate English revision performed by a native speaker would polish the text from typos\imperfections in the language still presnet along the manuscript.

3) Few questions are still open: what about uptake mechanism of such carriers? are they suitable for beeing scaled up into a translation to clinic or industry? Please comment\analyse these issues\points.

4) Furthermore, I would reccomend a thorough analysis of pitfalls and caveats related to the use of such nanocarriers by comparing them with properties and advantages vs drawbacks of other competitive nanocarriers.

Author Response

The manuscript entitled " ROS- and pH-Responsive Polydopamine Functionalized Ti3C2Tx MXene-Based Nanoparticles as Drug Delivery Nanocarriers with High Antibacterial Activity" authored by Wei-Jin Zhang and co-authors reports on functionalized MXene-derived nanoparticles as nanovectors for antibacterial drug delivery. They used several microscopic and spectroscopic  techniques to investigate chemical properties and to characterise their morphology\size\charges and so on. The paper is well structured and fulfils totally the aim and interest of Nanomaterials broad audience. I would recommend just minor changes\revisions in order to reach standard for publication In nanomaterials and to improve the quality of manuscript. Below there is a (short) list of suggestions\reccomendations\issues that authors should follow \address before re-submitting it:

-Many thanks for your positive comments on our manuscript. Here are our feedbacks on your specific comments.

Q1: I would suggest a comprehensive literature update by adding few research articles\reviews comparing them with competitive nanocarriers in the market\scientific literature.

Answer 1: Thank you very much for your kind comments. Based on your comments, we updated the Introduction with adding more discussion on the ROS levels in cancer cells and cargo efficiency on Page 2, L81-86 and Page 3, L100-116, respectively. In addition, the photothermal conversion efficiency and the extinction coefficient of the nanoplatform have been compared with the reported nanomaterials, as marked in red on Page 13, L 550-557, and Page 14L 593-599, respectively. For this revision, we added several more references as [22-25], [31-34], [67,68], [73]).  

Accordingly, all references were renumbered.

Q2: A moderate English revision performed by a native speaker would polish the text from typos\imperfections in the language still present along the manuscript.

Answer 2: Thank you for kind comment. Based on your comments, we have revised English thorough the text with the aid of a native speaker.

Q3: Few questions are still open: what about uptake mechanism of such carriers? are they suitable for being scaled up into a translation to clinic or industry? Please comment\analyse these issues\points.

Answer 3: Thank you very much for your valuable comments. We apologize for the lack of cellular uptake data for this work. Instead, we tried to explain the cellular uptake mechanism of MXene-based nanomaterials with referring to other reported literature, as marker in red on Page 3 L100-109. Furthermore, we added more explanation on the responses to clinical trials or industrialization of nanoparticles, as marked in red on Page 3, L109-116.

Two-dimensional transition metal carbides/carbonitrides known as MXenes are rapidly growing as multimodal nanoplatforms in biomedicine. From the carrier design concept, this nanoplatform is still very promising for application in the biomedical field. Firstly, the substrate matrix MXene is not only a high photothermal conversion material but also a biocompatible and non-toxic material, which is also ROS generator. Secondly, ROS is one of the unique hallmarks of tumor tissue. For instance, hypoxia dramatically changes ROS within cancer cells. ROS levels in tumor tissues (up to 100 × 10−6 M) are much higher than in normal tissue (≈ 20 × 10−9 M). Therefore, the ROS response is one of the most sensitive stimulus response modes that can effectively improve the tumor specificity of the vector. Final, polydopamine (PDA), a biomimetic material, not only has photothermal conversion capability and antibacterial activity but also can effectively improve carrier stability. Therefore, from the authors’ point of view, this nanoplatform is suitable for being scaled up and has clinical trials possibility. Though all of these points are fully discussed in the Introduction, we newly added more description on this matter to incorporate your comments in Page 3, L81-86 and Page 4, L100-116. May we ask your warm understanding?

Q4: Furthermore, I would recommend a thorough analysis of pitfalls and caveats related to the use of such nanocarriers by comparing them with properties and advantages vs drawbacks of other competitive nanocarriers.

Answer 4: Thank you very much for your valuable comments. The properties and advantages vs. drawbacks of nanomaterials have been compared with previously published works, as indicated in red on Pages 13, L550-557, Pages 14, L593-600, and Pages 3, L109-116, respectively. Furthermore, we also added more information about ROS-responsive, as marked in red on Page 2, L81-86. For this revision, we added several more references as [22-25], [31-34], [67,68], [73]).

Accordingly, all references were renumbered.

We thank you very much for your valuable comments and suggestions. Our revision is highlighted in red in this revised manuscript. We did our best to incorporate your valuable comments in this revised manuscript. We believe the quality of this revised manuscript would have been significantly improved. We hope that our revision could have been done successfully.

Reviewer 2 Report

The authors have prepared ROS-cleavable MXene-TK-DOX@PDA for effective chemotherapy drug delivery and antibacterial applications. The DOX of MXene-TK-DOX@PDA nanoparticles has revealed both ROS-responsive and pH-responsive release performance due to the cleavage of the thioketal linker. The MXene-TK-DOX@PDA nanoparticles have also displayed high antibacterial activity against both Gram-negative E. coli and Gram-positive B. subtilis within 5 h. Overall, this work can inspire more material design ideas of MXene-based nanoparticles for biomedical field application. Therefore, I would like to recommend this work to publish in Nanomaterials.

Herein, I have a suggestion for the authors. For the introduction “MXenes have been used in a variety of fields, including batteries [3-5], electromagnetic interference (EMI) shielding [6], energy storage [7], and catalysis [8].”, more references could be cited to broaden the introduction.

https://doi.org/10.2147/IJN.S328767

Author Response

The authors have prepared ROS-cleavable MXene-TK-DOX@PDA for effective chemotherapy drug delivery and antibacterial applications. The DOX of MXene-TK-DOX@PDA nanoparticles has revealed both ROS-responsive and pH-responsive release performance due to the cleavage of the thioketal linker. The MXene-TK-DOX@PDA nanoparticles have also displayed high antibacterial activity against both Gram-negative E. coli and Gram-positive B. subtilis within 5 h. Overall, this work can inspire more

material design ideas of MXene-based nanoparticles for biomedical field application.

Therefore, I would like to recommend this work to publish in Nanomaterials.

-Many thanks for your positive comments on our manuscript. Here is our feedback on your specific comments.

Q: Herein, I have a suggestion for the authors. For the introduction “MXenes have been used in a variety of fields, including batteries [3-5], electromagnetic interference (EMI) shielding [6], energy storage [7], and catalysis [8].”, more references could be cited to broaden the introduction.

https://doi.org/10.2147/IJN.S328767

Answer; Thank you very much for your kind words. As you recommended, we have cited the article as the new reference [8], as marked in red on Page 2, L49 (with adding the phrase, “antibacterial applications [8]”) and Page 21, L797–798.

Accordingly, all references were renumbered from the reference [9] to the end.

We thank you very much for your valuable comments and suggestions. Our revision is highlighted in red in this revised manuscript. We did our best to incorporate your valuable comments in this revised manuscript. We believe the quality of this revised manuscript would have been significantly improved. We hope that our revision could have been done successfully.
